# Modification and Expression of mRNA m6A in the Lateral Habenular of Rats after Long-Term Exposure to Blue Light during the Sleep Period

**DOI:** 10.3390/genes14010143

**Published:** 2023-01-04

**Authors:** Yinhan Li, Jinjin Ren, Zhaoting Zhang, Yali Weng, Jian Zhang, Xinhui Zou, Siying Wu, Hong Hu

**Affiliations:** 1Fujian Key Laboratory of Environmental Factors and Cancer, School of Public Health, Fujian Medical University, Fuzhou 350108, China; 2Department of Epidemiology and Health Statistics, School of Public Health, Fujian Medical University, Fuzhou 350108, China; 3Key Laboratory of Environment and Health, School of Public Health, Fujian Medical University, Fuzhou 350108, China; 4Department of Biochemistry and Molecular Biology, School of Basic Medical Sciences, Fujian Medical University, Fuzhou 350108, China; 5Key Laboratory of Ministry of Education for Gastrointestinal Cancer, School of Basic Medical Sciences, Fujian Medical University, Fuzhou 350108, China; 6School of Public Health, Fujian Medical University, Fuzhou 350108, China; 7Department of Preventive Medicine, School of Public Health, Fujian Medical University, Fuzhou 350108, China

**Keywords:** blue light, lateral habenula, m6A, depression, synaptic plasticity

## Abstract

Artificial lighting, especially blue light, is becoming a public-health risk. Excessive exposure to blue light at night has been reported to be associated with brain diseases. However, the mechanisms underlying neuropathy induced by blue light remain unclear. An early anatomical tracing study described the projection of the retina to the lateral habenula (LHb), whereas more mechanistic reports are available on multiple brain functions and neuropsychiatric disorders in the LHb, which are rarely seen in epigenetic studies, particularly N6-methyladenosine (m6A). The purpose of our study was to first expose Sprague-Dawley rats to blue light (6.11 ± 0.05 mW/cm^2^, the same irradiance as 200 lx of white light in the control group) for 4 h, and simultaneously provide white light to the control group for the same time to enter a sleep period. The experiment was conducted over 12 weeks. RNA m6A modifications and different mRNA transcriptome profiles were observed in the LHb. We refer to this experimental group as BLS. High-throughput MeRIP-seq and mRNA-seq were performed, and we used bioinformatics to analyze the data. There were 188 genes in the LHb that overlapped between differentially m6A-modified mRNA and differentially expressed mRNA. The Kyoto Encyclopedia of Genes and Genomes and gene ontology analysis were used to enrich neuroactive ligand–receptor interaction, long-term depression, the cyclic guanosine monophosphate-dependent protein kinase G (cGMP-PKG) signaling pathway, and circadian entrainment. The m6A methylation level of the target genes in the BLS group was disordered. In conclusion, this study suggests that the mRNA expression and their m6A of the LHb were abnormal after blue light exposure during the sleep period, and the methylation levels of target genes related to synaptic plasticity were disturbed. This study offers a theoretical basis for the scientific use of light.

## 1. Introduction

Humans and other creatures have adapted to a consistent and predictable 24 h solar cycle [1]. However, with the wide application of artificial light sources with high blue light abundance, such as light-emitting diodes (LED), mobile phones, and computers, light-at-night (LAN) has become increasingly common. LAN may cause dysregulation of physiological functions such as biological rhythm, sleep and arousal, cognition, and mood [2,3].

It has previously been reported that long-term and high-intensity blue light can induce genotoxic stress in cells [4]. In addition, exposure to low illumination (less than 1000 lx) blue light for 4 h stimulates the growth speed of the eyes of chicks and affects the ocular rhythm, which indicates that such exposure may be deleterious to emmetropization in children [5]. Previous studies have shown that excessive exposure to LAN can increase the risk of depression [1,2,6]. The intrinsically photosensitive retinal ganglion cells (ipRGCs) in the retina are involved in the regulation of serum melatonin levels, sleep, and biological circadian rhythms [7,8]. The ipRGCs are most sensitive to blue light [9,10]. A recent study showed that blue light at night can induce depression in mice via the circadian rhythm-controlled subcortical intrinsically photosensitive retinal ganglion cell-habenula dorsal-nucleus accumbens (ipRGC-dpHb-NAc) pathway [11]. However, the mechanisms by which blue light affects the central nervous system are still poorly understood.

According to the coordinates of the Montreal Neurological Institute (MNI), the LHb is located in the posterior parietal thalamus (PPtha) [12] and mainly carries glutamate neurons that pass the GABAergic relay in the nodular tegmental nucleus, which inhibits the reward system of the brain by connecting with the interneurons in the ventral tegmental area (VTA) and the dorsal raphe nucleus (DRN) [13]. It is reported to be involved in various brain functions and neuropsychiatric disorders, such as drug abuse, reward aversion, pain, sleep, and other pathophysiological changes in mental diseases, especially severe depression [14]. Recently, the LHb, the brain’s “anti-reward system”, was shown to play an important role in various molecular and electrophysiological characteristics in depression. It was found that the new antidepressant ketamine inhibits the downstream monoaminergic reward center by blocking the N-methyl-D-aspartate receptor (NMDAR) dependent burst activity of the LHb [15]. Kir4.1 in LHb astrocytes is up-regulated in depressed mice [16], and regulation of CB_1_R in LHb astrocytes may help regulate neuronal and synaptic activity as a way to regulate mood and improve depressive symptoms [17]. However, little is known about the epigenetic mechanisms of the LHb.

Recent studies showed a close relationship between epigenetics and the molecular mechanisms of depression. At present, epigenetic modifications related to depression include DNA methylation, post-translational histone modifications, and microRNAs (miRNAs) or long non-coding RNAs (lncRNAs) [18]. In addition, some studies showed that methylation of the fifth carbon atom of cytosine in the heterocyclic aromatic ring produces 5-methylcytosine (5mC) [19]. In addition, epigenetic modifications of RNA also include N1 methyladenosine (m1A) [20], N6-methyladenosine (m6A) [21], and 7-methylguanine nucleoside (m7G) [22], which affect various protein functions that regulate disease development [23].

Desrosiers et al. (1975) first proposed a new RNA epigenetic modification, N6-methyladenosine (m6A); m6A-related regulatory proteins include methyltransferases (“writers”), demethylases (“erasers”), and methylated reading proteins (“readers”) [24]. The modification of m6A is co-catalyzed by methyltransferases composed of METTL3, METTL14, and WTAP, and then preferentially combined with YTHDF and IGFBP to participate in the translation and degradation of downstream RNA [25]. The m6A modification of RNA has also proved to be reversible because it is bi-directionally regulated by m6A methyltransferase and demethylase (including ALKBH5 and FTO) [26]. m6A-related proteins are expressed in almost all cells. In the nervous system, m6A is involved in neurogenesis, brain capacity, learning and memory, memory formation, and consolidation, and is related to the development of Parkinson’s disease, Alzheimer’s disease, multiple sclerosis, depression, epilepsy, brain tumors, and other diseases [23]. Research shows that 23 m6A modifying genes were genotyped in healthy individuals and patients with major depressive disorder (MDD), and found rs12936694 within the ALKBH5 region to be significantly associated with MDD [27]. Additional studies found that the RNA demethylase FTO, one of the genes associated with depression, was significantly downregulated in the serum of depressed patients and the hippocampus of mice with depression-like behaviors [28].

In this study, we hypothesized that the genes in the LHb of rats underwent m6A methylation modification after 12 weeks of blue light exposure during their sleep period. We further analyzed the data using high-throughput m6A MeRIP-seq and mRNA-seq to determine the potential epigenetic molecular mechanisms.

## 2. Materials and Methods

### 2.1. Animals

Male Sprague-Dawley (SD) rats weighing 74.3 ± 0.78 g (3 weeks) were obtained from the Experimental Animal Center of Fujian Medical University. Two rats in each cage were placed in transparent (polycarbonate) cages with controlled temperatures (23 ± 1 °C), relative humidity 50–70%, and controlled noise (less than 60 dB (A)). The light was set to a 10 h light/14 h dark cycle. The rats were free to obtain standardized granular food and tap water.

### 2.2. Blue Light Treatment

An LED light source was used in the experiment, and the light spectrum was measured (Figure 1A,B). After five days of adaptation, 36 male SD rats were randomly divided into control and BLS groups according to body weight. (*n* = 18/group). According to the requirements for the environment and housing facilities of experimental animals (GB 149,252,010, China), the cage containing the control subjects was raised from 10:00 to 20:00 under a 10 h light/14 h dark cycle (200 L × white light). BLS was supplemented with blue light for 4 h from 6:00 to 10:00 before it was given white light, and the irradiance of blue light was the same as that of the control group to exclude the influence of different light energies. The illumination intensity and irradiance were detected using a spectral irradiance colorimeter (Everfine, Hangzhou, China). The lighting intensity during the dark period was 0 Lx. At the 12th week of the experiment, the rats were anesthetized with isoflurane, and we collected serum and LHb (Figure 1C). All treatments were performed under mild care, and mice suffering was minimized. The Institutional Animal Care and Use Committee of Fujian Medical University approved all experiments in the present study.

### 2.3. RNA Preparation

Total RNA was extracted by adding TRIzol reagent (Invitrogen, Waltham, MA, USA; cat. no. 15,596,026) to the LHb. RNA quality was determined by testing A260/A280 using a Nanodrop^TM^ OneC spectrophotometer (Thermo Fisher Scientific Inc., Waltham, MA, USA). RNA integrity was confirmed using 1.5% agarose gel electrophoresis. Qualified RNA was quantitated with Qubit3.0 and the Qubit^TM^ RNA Broad Range Assay kit (Life Technologies, Thermo Fisher Scientific Inc., Waltham, MA, USA; Q10210).

### 2.4. High-Throughput m6A

MeRIP experiments, high-throughput sequencing, and data analysis were conducted by SeqHealth Technology Co., Ltd. (Wuhan, China). First, we used 50 μg of total RNA-enriched polyadenylate RNA through VAHTS mRNA Capture Beads (VAHTS, Nanjing, China; cat. no. N401-01/02). Then, 20 mM ZnCl_2_ was added to the mRNA and incubated at 95 °C for 5–10 min until the RNA fragments were mainly distributed in 100–200 nt. Subsequently, 10% of the RNA fragments were stored as “inputs”, and the rest were used for m6A immunoprecipitation (IP). A specific anti-m6A antibody (Synaptic Systems, Göttingen, Germany; 202,203) was used for m6A IP. TRIzol reagent (cat. no. 15,596,026) to prepare RNA samples for input and IP. Finally, according to the manufacturer’s instructions, the KC-Digital^TM^ Stranded mRNA Library Prep Kit for Illumina^®^ (Wuhan SeqHealth Co., Ltd., Wuhan, China; cat. no. DR08502) was used to construct the chain mRNA. This kit eliminates duplication bias in the PCR and sequencing steps by labeling pre-amplified cDNA molecules with a unique molecular identifier (UMI) of eight random bases. Library products corresponding to 200–500 base points were enriched, quantified, and sequenced on a DNBSEQ-T7 sequencer (MGI Tech Co., Ltd., Shenzen, China) using the PE150 model.

### 2.5. Sequencing Data Analysis

Raw data were processed using Trim Galore (Cambridge, UK). StringTie (Baltimore, MD, USA) [29] was used to analyze the RNA expression levels, and DESeq was used to calculate the differential expression [30]. Exomepeak2 (Suzhou, China) [31,32] was used for m6A peak calling and differential methylation detection, and a Poisson generalized linear model was used to estimate methylation levels and detect differential methylation regions. ExomePeak2 estimates the size factor of the sequencing depth in the non-methylated background area. The consensus m6A motif sequences were identified using STREME [33]. The STREME algorithm integrated the position weight matrix Markov model to report a useful estimate of the statistical significance of each discovered motif. The annotation of KEGG and GO was completed using KOBAS (Beijing, China) and DAVID (Frederick, MD, USA) [34,35], respectively. DirectRMDB (Suzhou, China) [36] was used to analyze the post-transcriptional RNA modifications. Geo2vec (Suzhou, China) [37] was used to analyze them6A prediction models based on geographic information. The m6A methylation sites were obtained from m6A-TSHub (Suzhou, China) [38]. The pitranscriptome analysis was performed by MetaTX (Suzhou, China) [39]. The m6A regulator substrate identified using CLIP technology can be downloaded from the POSTAR3 [40] and ENCORI [41] databases (Guangzhou, China).

### 2.6. Western Blotting Assay

Western blotting was performed as previously described [42]. Proteins were extracted using RIPA Lysis Buffer and denatured with 5X protein loading buffer. SDS-PAGE was performed using 12% running gels, and the resolved proteins were transferred onto PVDF membranes. The PVDF membranes were blocked with non-fat milk for 1 h and incubated with FTO (1:1000), ALKBH5 (1:1000), METTL3 (1:1000), and YTHDC2 (1:1000) antibodies at 4 °C overnight. Next, the secondary antibodies covered the membranes for 1 h at 37 °C. The grayscale of the protein bands was analyzed using ImageJ (NationalInstitutes of Health, USA) software.

### 2.7. Enzyme-Linked Immunosorbent Assays for Serum Determinations

Serum melatonin levels were determined according to the manufacturer’s instructions (Elabscience, Wuhan, China). Briefly, 50 mL of standard or sample was added to each well, and then 50 mL of biotinylated detection antibody was immediately added to each well. Next, the plate was incubated for 45 min at 37 °C and washed three times. Then, 100 mL of HRP conjugate was added to each well, and the plate was incubated for 30 min at 37 °C and washed five times. Then, 90 mL of substrate reagent was added, and the plate was incubated for 15 min at 37 °C. Finally, 50 mL of Stop Solution was added, and the absorbance at 450 nm was immediately determined. Subsequently, the results were analyzed.

### 2.8. Molecular Docking

To explore whether melatonin can interact with m6A-modified differential expression regulators, SYBYL-X 2.0 software (Tripos, St. Louis, MO, USA) was used for molecular docking. SYBYL-X 2.0 was used to prepare protein structures to remove water molecules and heteroatoms, add hydrogen atoms, and repair side chains [43]. The 2D structure of melatonin was downloaded from PubChem. The total score, which is a comprehensive evaluation of hydrophobic complementarity, polar complementarity, solvation terms, and entropic terms, was considered a stable interaction when the value was higher than 5 [44].

### 2.9. Statistical Analysis

Results are presented as mean ± standard deviation (SD). Unpaired Student’s *t*-tests were conducted using SPSS 25. A *p*-value of <0.05 was assumed to be statistically significant.

## 3. Results

### 3.1. Transcriptome-Wide Detection of m6A Modification in LHb

To explore the mechanism of blue light exposure in SD rats during the sleep period in the lateral habenular nucleus, MeRIP-seq and RNA-seq analyses were performed. In BLS, the R package exomePeak identified 10,561 m6A peaks containing 21,039 gene transcripts. Similarly, 10,708 m6A peaks were found in the control group, representing 24,458 transcripts of genes. In addition, 10,140 peaks were found at the intersection of the BLS and control groups, corresponding to 15,342 genes. After crossing the normal m6A peak in the rat brain, 232 genes in the BLS and control groups did not undergo m6A methylation changes (Figure 2A,B). Most genes had 1–3 m6A methylation peaks, whereas in the BLS and control groups, relatively few genes had four or more m6A methylation peaks (Figure 2C). We used Streme to determine the presence of an m6A consensus sequence for RRACH (where R represents a purine, A is m6A, and H is a non-guanine base) reported for the detection of m6A (Figure 2D).

### 3.2. Distribution of m6A Modification in Transcriptome

The distribution of m6A methylation in the whole transcriptome of the BLS and control groups was analyzed. The results showed that m6A modification was enriched in the 5′untranslated region (5’UTR), starting codon, coding sequence (CDS), ending codon, and 3′UTR. The m6A peak density increased rapidly between the 5 ′UTR and the starting codon and was relatively flat in the CDS region. The highest-density region existed near the stop codon. In the 3′UTR region, the density of the m6A peak decreased rapidly, and the number of m6A peaks was also very high (Figure 3).

### 3.3. Differentially Methylated Genes and Differentially Expressed Genes

We set the statistical standard of differential methylation and differential expression genes as *p* ≤ 0.05. A total of 4171 differentially expressed mRNA modified by m6A were identified through m6A sequence data. RNA sequence analysis revealed that 585 mRNA were differentially expressed between the BLS and control groups. In addition, 188 genes were observed in the overlap of differentially m6A-modified mRNA and differentially expressed mRNA (Figure 4A), 41 of which were upregulated and 64 were downregulated (Figure 4B).

### 3.4. KEGG and GO Annotation of the Overlap of Differentially m6A-Modified mRNA and Differentially Expressed mRNA

Using KEGG pathway and GO enrichment analysis on the DAVID web server, 188 overlapping genes of differentially m6A-modified mRNA and differentially expressed mRNA were related to important pathways and biological functions. KEGG pathway analysis showed that these genes were mainly involved in neuroactive ligand–receptor interactions, long-term depression, the cGMP-PKG signaling pathway, and circadian entrainment (Figure 5). GO enrichment analysis can be divided into three functional categories: molecular function (MF), cellular component (CC), and biological process (BP). MF terms included calcium ion binding and phosphatidylserine binding. The CC term included the regulation of excitatory synapses, integral components of the postsynaptic density membranes, and GABAergic synapses. BP terms included regulation of presynaptic assembly, neuron projection morphogenesis, and nervous system development (Figure 6).

### 3.5. Neuroactive Ligand Receptor Interaction, Circadian Rhythm Entrainment, cGMP-PKG Signal Pathway and Regulation of Potential Regulators

According to the RNA sequence data of LHb, there were significant differences in mRNA expression levels of genes related to neuroactive ligand–receptor interaction (DRD2, NTS, CCK, LEPR, GRIN2A), circadian entrainment (CACNA1C, PLCB4, GUCY1A1, CACNA1I), and the cGMP-PKG signaling pathway (PDE2A, ADRA1B, ATP2B1, ADRA2C). CLIP technology was used to search the POSTAR3 and ENCORI databases, and IGF2BP1, a reader, was found to participate in the regulation of most of these genes (Table 1, Figure 7A–C).

### 3.6. Potential RNA m6A Regulators of Different Methylation Genes

To identify potential regulators of RNA m6A methylation, we analyzed the mRNA expression levels of 19 RNA m6A methylation writers, readers, and erasers. As shown in Table 2, there was no difference in the *p*-values. We performed western blot analysis on METTL3, FTO, ALKBH5, and YTHDC2 and found that METTL3 (*p* = 0.148639; Figure 8A) was not significantly different from the control group, but FTO (*p* = 0.017; Figure 8B) and ALKBH5 (*p* = 0.025; Figure 8D) were up-regulated (Figure 8B), and YTHDC2 (*p* = 0.049; Figure 8D) was downregulated.

### 3.7. Molecular Interactions of Melatonin with Differently Expressed Regulators of RNA m6A Modification

Theoretically, the melatonin content in the BLS was significantly lower than that in the control group, with a statistically significant difference (*p* = 0.001124; Figure 9A). We evaluated the molecular interactions of melatonin with differentially expressed regulators of m6A modification, including METTL5, FTO, ALKBH5, YTHDF2, YTHDF3, IGF2BP2, and FMR1. The results are shown in Table 3 and Figure 9B–H. Melatonin bound to METTL5 via the formation of one hydrogen bond at Met104 and 12 hydrophobic contacts with Ile102, Amn0, Asp103, Lys2, Thr5, Met116, Leu98, Leu9, Leu23, Val105, Pro114, and Gly112 (Figure 9B); FTO via the formation of five hydrogen bonds at Leu435, Va1421, Leu496, Ile492, and Leu439, and five hydrophobic contacts with Arg431, Gln499, Leu500, Leu439, and Ile492 (Figure 9C); ALKBH5 via the formation of one hydrogen bond at Arg1102 and one hydrophobic contact with Trp1007 (Figure 9D); YTHDF3 via the formation of one hydrogen bond at Lys414 (Figure 9E); YTHDF2 via the formation of two hydrogen bonds at Arg447 and Ala444 and one hydrophobic contact with Ser448 (Figure 9F); IGF2BP2 via the formation of three hydrogen bonds at Ile55, Ile52, and Tyr73, and one hydrophobic contact with Ala51 (Figure 9G); and FMR1 via the formation of two hydrogen bonds at Leu54 and Ser49, and seven hydrophobic contacts with Glu50, Asp48, Ser52, Ile55, Pro46, His21, and Lys19 (Figure 9H).

## 4. Discussion

Epidemiology and animal experiments have shown that overexposure to visible light in the high-energy blue light band leads to circadian rhythm disorder. This leads to sleep disorders and induces various negative emotions, resulting in various physiological and functional diseases [45,46]. When melanopsin, which is most sensitive to blue light in ipRGCs, receives a light signal from the retina, it sends the light information to the suprachiasmatic nucleus (SCN) and other structures in the brain, including the LHb. Some studies have confirmed that LHb neurons respond to retinal light using multi-electrode probes that simultaneously record the multilevel activity of the LHb [47]. However, the molecular mechanisms of the LHb after blue light exposure remain largely unknown.

Epigenetic studies have demonstrated a common genetic basis for mental diseases. In this study, rats that entered the sleep period were exposed to 4 h of blue light for 12 weeks, and then the LHb was extracted. Based on the high-throughput m6A MeRIP and mRNA sequences, our research results first identified the potential epigenetic changes in m6A modification and mRNA transcription profiles in the LHb exposed to blue light.

In this study, GO and KEGG pathway enrichment analyses were performed to explore the biological functional changes caused by mRNA m6A modification. A total of 188 genes overlapped with the differentially m6A-modified mRNAs and differentially expressed mRNAs. The top 20 KEGG terms included cGMP-PKG signaling pathway, calcium signaling pathway, long-term depression, neuroactive ligand receptor interaction, and circadian rhythm. In GO terms, calcium ion binding and metal ion binding were enriched in MF. Excitatory synapses, a component of the postsynaptic density membrane, are abundant in the CC, as are GABAergic synapses, regulation of presynaptic assembly in the BP, and projection neuron morphogenesis, which is enriched in nervous system development. Previous studies have shown that the occurrence of depression is closely related to changes in synaptic plasticity. Most antidepressants modulate synaptic transmission by affecting the levels of monoamine neurotransmitters secreted at the synapse and by using the neurotransmitter glutamate, suggesting that antidepressants can modulate synaptic plasticity [48]. In addition, the recently discovered antidepressant properties of ketamine work through sustained potentiation of excitatory synapses [49]. More studies have found that a single exposure to a stressor facilitates the induction of LTP in the LHb, suggesting that animal models of depression or post-traumatic stress disorder have altered synaptic plasticity in the LHb [50].

Through the analysis of m6A MeRIP sequence data, the distribution of m6A modification peaks in the control and BLS groups showed that most genes had 1–3 peaks and parts. When the gene was divided into the 5′ UTR, start codon, CDS, stop codon, and 3′ UTR, most m6A modification peaks were located in the CDS and 3’ UTR regions, and the highest value was near the stop codon. The 3′ UTR was shown to regulate different protein characteristics, including the formation of protein complexes or post-translational modifications [51]. Therefore, the increase in the number of m6A modification sites at the termination codon and 3′-UTR region may be related to mRNA stability and translation.

The biological effects of RNA m6A modification depend on the recognition and binding of m6A binding proteins [52]. Using the computational prediction method, we found no difference in the mRNA of m6A related methylation regulatory proteins in the LHb. However, we measured the protein content of METTL3, FTO, ALKBH5, and YTHDC2 and found that the results were different from the mRNA expression. Methylated regulatory proteins may change during translation because tRNAs also interact with ribosomes to play a central role in translation; thus, the abundance, availability, and codon usage of tRNAs in mRNAs have been reported to strongly affect the rate and efficiency of translation [53]. Studies have shown that IGF2BP1 is located in the axon, dendritic spine, and neuronal cell bodies, enabling mRNA 3′-UTR binding activity [54]. It is also involved in RNA localization [54] and dendritic dendronization [55,56]. Our results showed that IGF2BP1 is involved in the m6A modification of relevant differentially expressed genes in this pathway. For example, GUCY1A1 (guanylate cyclase 1 (1) is a soluble guanylate-circulating enzyme. It was found that GUCY1A1 can regulate the release of glutamate and GABA in the somatosensory cortex of mice through nitric oxide/cGMP signals [57]. In our sequencing results, we found that after BLS exposure, the methylation level of GUCY1A1 mRNA m6A in the LHb was low, and the stability and translation level of GUCY1A1 mRNA changed after the recognition of GUCY1A1 mRNA by IGF2BP1, which may affect the release of glutamate and GABA, thus affecting synaptic function.

Since blue light is the most effective light to inhibit melatonin [58], we measured the content of melatonin in the serum of rats in the two groups and found that melatonin in the BLS group was significantly lower than that in the control group. We combined melatonin with m6A regulatory factors, among which METTL3, FTO, ALKBH5, and YTHDCF2 had total scores higher than 5. The lower melatonin levels in the BLS might be caused by METTL3, FTO, ALKBH5, and YTHDF2.

## 5. Conclusions

In conclusion, our data confirmed that the level of m6A methylation in the LHb changed when exposed to blue light. The sequencing results showed that blue light altered the expression of genes related to synaptic function in the lateral habenular nucleus. Our results showed that m6A modification might play a functional role in depression caused by synaptic dysfunction following blue light exposure (Figure 10; figdraw ID: SPTUPb1c0c). These findings provide a better understanding of the epigenetic mechanisms of the effects of blue light in the neural system.

## Figures and Tables

**Figure 1 genes-14-00143-f001:**
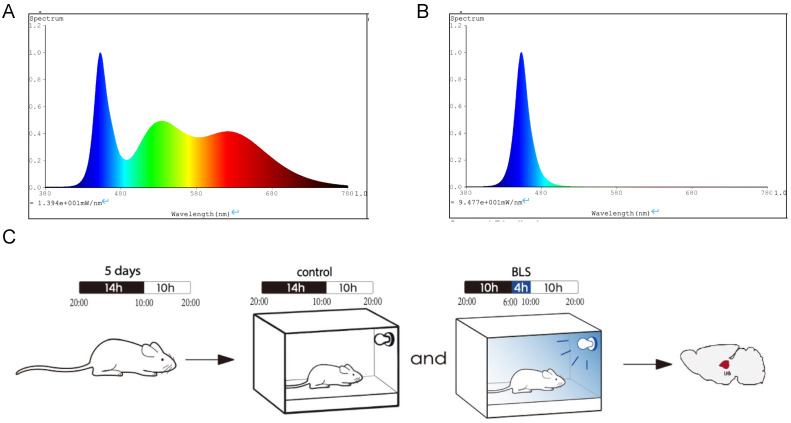
(**A**) White light spectrum. (**B**) Blue light spectrum. (**C**) Experimental design for control or BLS paradigms.

**Figure 2 genes-14-00143-f002:**
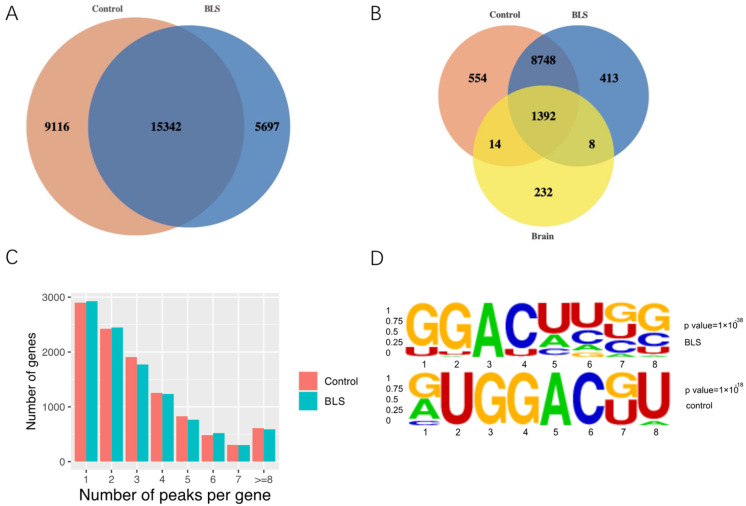
The m6A modification patterns in BLS and control groups. (**A**) The Venn diagram shows the overlap of two groups of m6A genes. (**B**) Venn diagram shows the overlap of m6A peak in the brain of two groups and normal rats. (**C**) The distribution of m6A methylation peaks in each gene. (**D**) The motifs for m6A peak regions based on STREME.

**Figure 3 genes-14-00143-f003:**
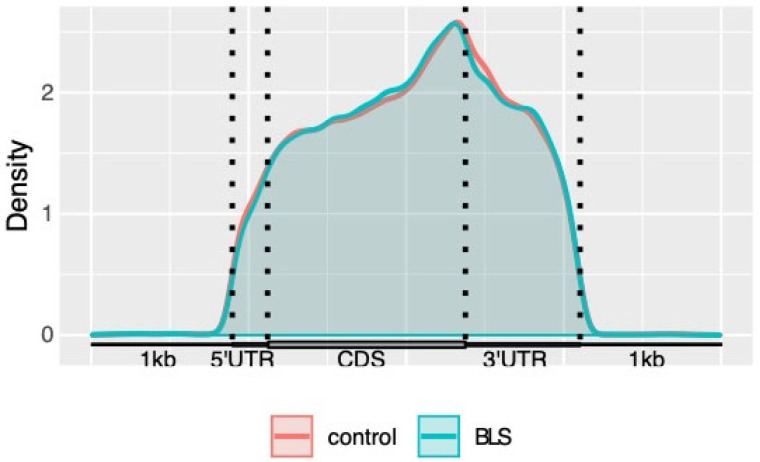
Density of m6A methylation peaks in mRNA transcripts.

**Figure 4 genes-14-00143-f004:**
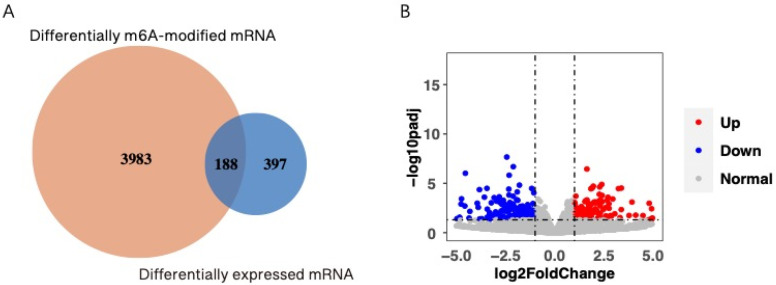
(**A**) Differences in m6A-modified mRNA and mRNA expression between BLS and control groups. (**B**) Volcanic map of mRNA differentially expressed between the two groups.

**Figure 5 genes-14-00143-f005:**
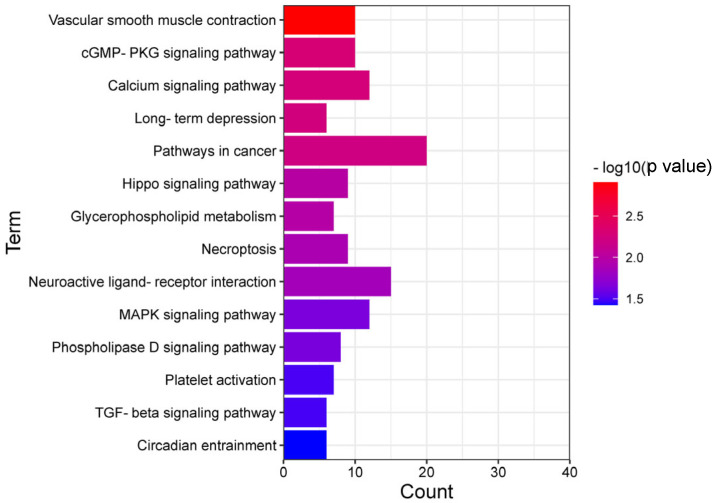
The top 20 KEGG analysis enriched pathways of the 188 overlapped genes of differentially m6A-modified mRNA and differentially expressed mRNA.

**Figure 6 genes-14-00143-f006:**
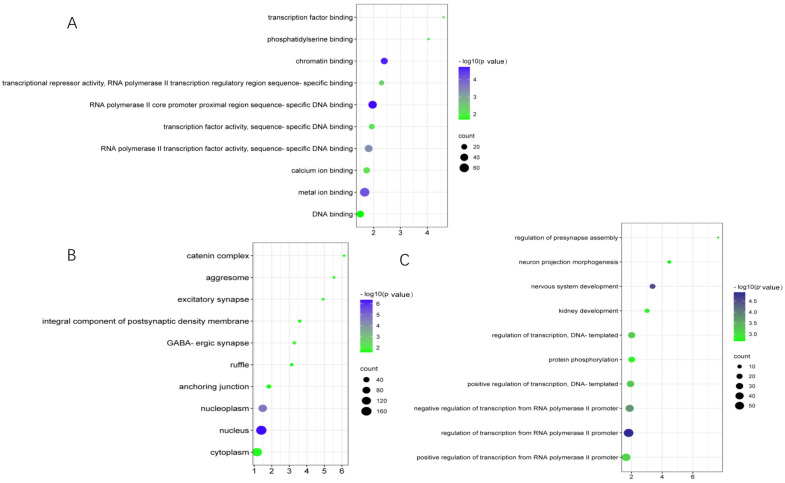
GO functional annotation of 188 overlapped genes of differentially m6A-modified mRNA and differentially expressed mRNA. (**A**) GO molecular function. (**B**) GO cellular component. (**C**) GO biological process.

**Figure 7 genes-14-00143-f007:**
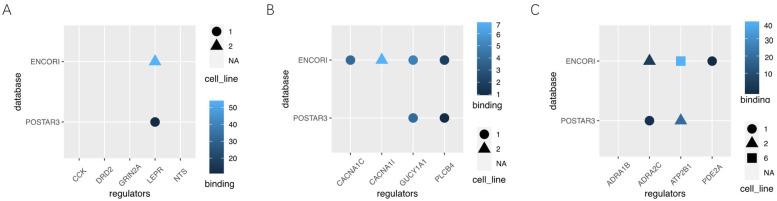
(**A**–**C**) The effects of IGF2BP1 on the differentially expressed genes of neuroactive ligand–receptor interaction, circadian entrainment, and cGMP-PKG signal pathway.

**Figure 8 genes-14-00143-f008:**
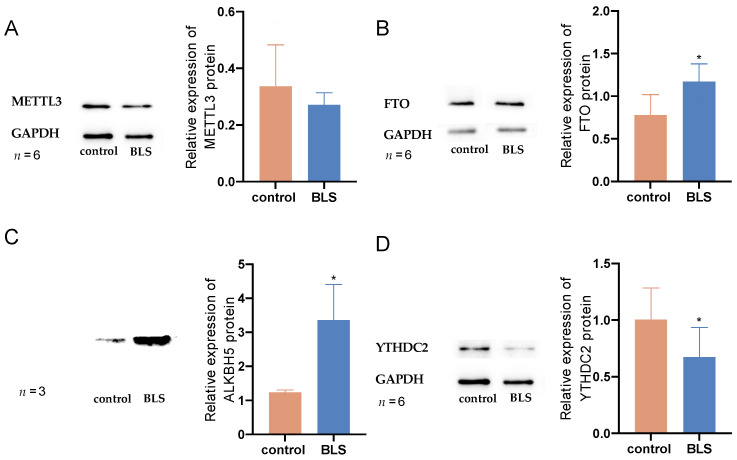
The protein expression level of m6A regulatory factor in LHb. (**A**–**D**) the protein levels of METTL3, FTO, ALKBH5, and YTHDC2 (* *p* < 0.05).

**Figure 9 genes-14-00143-f009:**
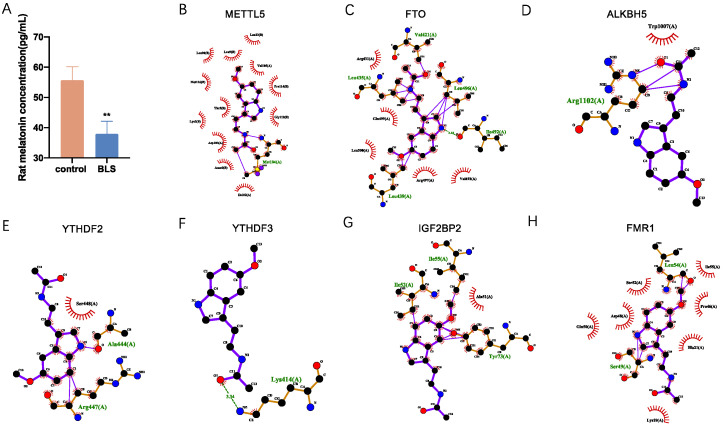
The secretion of melatonin and the molecular interaction between different expression regulators modified with m6A and carnitine. A two-dimensional interaction model is proposed. (**A**) Melatonin contents, (**B**) METTL3, (**C**) FTO, (**D**) ALKBH5, (**E**) YTHDF2, (**F**) YTHDF3, (**G**) IGF2BP2, and (**H**) FMR1. (** *p* < 0.01).

**Figure 10 genes-14-00143-f010:**
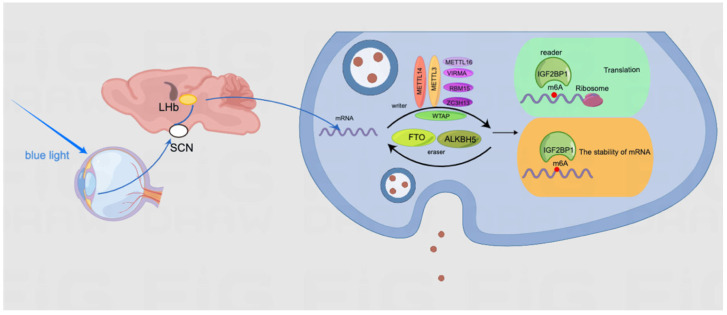
Blue light is delivered to the LHb via the SCN pacemaker after it is accepted by melanopsin in the retina, where m6A methylation of genes related to synaptic function is abnormal. Therefore, blue light exposure may disrupt the translation process of related genes as well as mRNA stability through m6A modification processes.

**Table 1 genes-14-00143-t001:** The level of protein interaction and mRNA interaction between IGF2BP1 and some related differential genes in the neuroactive ligand–receptor interaction, circadian entrainment, and cGMP-PKG signal pathway.

	Gene	log2FoldChange	*p* Value	IGF2BP1	
				POSTAR3	ENCORI
Neuroactive ligand-receptor interaction	*DRD21*	2.410717348	0.000718739	0	0
	*NTS*	2.289483776	0.000814387	0	0
	*CCK*	−3.891976901	0.002732121	0	0
	*LEPR*	−1.544900237	0.018075531	1	1
	*GRIN2A*	−1.383616013	0.048896803	0	0
Circadian entrainment	*CACNA1C*	−1.325997265	0.002641267	0	1
	*PLCB4*	−1.769449948	0.003986134	1	1
	*GUCY1A1*	−.359242738	0.032953756	1	1
	*CACNA1I*	0.077654698	0.85792273	0	1
cGMP-PKG signaling pathway	*PDE2A*	1.63634444	0.000483644	0	1
	*ADRA1B*	−2.789275777	0.000878733	0	0
	*ATP2B1*	−1.753044611	0.006740132	1	1
	*ADRA2C*	1.717512915	0.017494168	1	1

1 means that it interacts with IGF2BP1, and 0 means not.

**Table 2 genes-14-00143-t002:** mRNA expression level of m6A regulatory factor in LHb.

Gene	Regulation	Base Mean	log2FoldChange	*p* Value
*ALKBH5*	eraser	341.9384302	−0.109170657	0.771338043
*CBLL1*	writer	346.9551446	0.116591986	0.640932024
*FMR1*	reader	1847.917149	−0.071575928	0.756365327
*FTO*	eraser	1766.661301	0.156834515	0.728325104
*HNRNPA2B1*	reader	23816.12242	0.04414699	0.831082269
*HNRNPC*	reader	5475.18213	0.188188685	0.223239932
*IGF2BP1*	reader	1.168447989	3.574733353	0.364559468
*IGF2BP2*	reader	53.11726013	0.147512464	0.760550998
*IGF2BP3*	reader	8.376292315	0.613618984	0.540055918
*METTL14*	writer	1787.917122	0.116946577	0.522853135
*METTL3*	writer	2373.613619	−0.098001527	0.667618896
*METTL5*	writer	1275.304982	−0.179998413	0.419116968
*VIRMA*	writer	1113.272801	−0.226631023	0.341256528
*WTAP*	writer	1863.746967	0.106961708	0.735133477
*YTHDC1*	reader	7452.265261	−0.077633436	0.679592938
*YTHDF1*	reader	1043.115245	−0.087632591	0.633796561
*YTHDF2*	reader	1082.137679	0.153862012	0.481990904
*YTHDF3*	reader	732.4864141	0.064147425	0.865441311
*ZC3H13*	writer	13329.50949	−0.284321826	0.402424357

**Table 3 genes-14-00143-t003:** Molecular interactions between melatonin and m6A modified different expression regulators.

Protein	Regulation	Total Score	H-Bond Number	Residues Involved in H-Bond Formation	Hydrophobic Contacts Number	Residues Involved in Hydrophobic Contacts
METTL5	writer	10	1	Met104	12	Ile102, Amn0, Asp103, Lys2, Thr5, Met116, Leu98, Leu9, Leu23, Val105, Pro114, Gly112
FTO	eraser	9.13	5	Leu435, Va1421, Leu496, Ile492, Leu439	5	Arg431, Gln499, Leu500, Leu439, Ile492
ALKBH5	eraser	4.39	1	Arg1102	1	Trp1007
YTHDF3	reader	8.152	1	Lys414	0	
YTHDF2	reader	6.401	2	Arg447, Ala444	1	Ser448
IGF2BP2	reader	4.891	3	Ile55, Ile52, Tyr73	1	Ala51
FMR1	reader	3.19	2	Leu54, Ser49	7	Glu50, Asp48, Ser52, Ile55, Pro46, His21, Lys19

## Data Availability

All relevant data are provided in the manuscript. Please contact huhong1009@fjmu.edu.cn for any raw data files and further analysis.

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
