# Peer review of "Modification and Expression of mRNA m6A in the Lateral Habenular of Rats after Long-Term Exposure to Blue Light during the Sleep Period"

_genes, 2023, doi:10.3390/genes14010143_

Round 1
Reviewer 1 Report
Remarks concerning the manuscript entitled: “Modification and Expression of mRNA m6A in Lateral Habenular of Rats After Long-term Exposure to Blue Light During Sleep Time.” By Yinhan Li et al.
Introduction
The object of the study was to investigate RNA m6A modification and different mRNA transcriptome profiles in the lateral habenular (LHb)of Sprague-Dawley (SD) rats after 12 weeks of blue light exposure during their sleep time (BLS). In their conclusion, the authors suggest that the mRNA expression and their m6A in LHb become abnormal after blue light exposure during the sleep time (BLS), and that the methylation level of target genes related to synaptic plasticity is disturbed. Their research is supposed to offer a scientific use of the light.
In a first instance, the manuscript seems to reflect an important piece of work, figures seem adequate as well as the referencing. However, there are too many drawbacks in various aspects of the manuscript rendering it difficult to read, difficult to understand. It is thus a hard task to achieve an objective review of the manuscript submitted.
Specific Remarks
1- Starting with the English form of the manuscript: (a) - In the abstract, line 22, “a public health hazard,” it is more adequate to employ a “public health risk.” (b) - Line 23, “with nervous diseases,” nervous is not adequate, perhaps more correct to employ, “brain diseases.” (c) - Line 24, “The mechanism about the effect of blue light exposure on the health is not inadequate”: it is difficult to understand what the authors mean. (d) - Lines 29-30, “Sprague-Dawley (SD) rats after 12 weeks of blue light exposure during their sleep time (BLS)”: the sentence is not precise enough, the authors submitted the animals to blue light (200 lux) 4h/day for 12 weeks. (e) - The authors claim in their summary that the blue light was applied during the light period, i.e., when the animals are sleeping. This is not in agreement with the material and method statement (see later). (f) - Moreover, the terminology employed, i.e. “the sleeping time” is also not adequate. Rats sleep either during the light or during the dark periods. But they sleep more during the light period/the dark period. (g) - Finally, line 35, ”In conclusion, this study suggested…”: It is more adequate to write, “This study suggests.”
2 – In the introduction the same difficulties are encountered: (a) - lines 58-59: “However, the mechanisms of blue light on the nervous system are still insufficient,” this sentence must be changed. We suppose that the authors want to say that the mechanisms of blue light on the central nervous system are still poorly understood. (b) - Again, line 160: “after 12 weeks of blue light exposure during their sleep time,” the terminology sleep time is not adequate.
3 – Material and methods subheading is really difficult for us. We give here examples: (a) - Lines 119-120, “… group according to body weight…”, what does this sentence mean? (b) - Moreover, the schedule employed is absolutely unclear (difficult for us to understand), i.e. “According to the requirements for the environment and housing facilities of experimental animals (GB 149252010, China), the cage containing the control subjects was raised in a 10:00 to 20:00, 10 h light/ 14 h dark cycle (~200 lx)”. According to this protocol, we understand that the animals were in light-on from 10h to 20h (10h in light-on), then after, they were in light-off during the remaining 14h of the day, i.e., from 20h to 10h (14h in light-off). (c) - In the following sentence, however, lines 122-124, it is written: “BLS supplemented blue light for 4 hours from 6:00 to 10:00….. “ i.e., during the dark phase of the 24h period. At this step, we are lost. We cannot understand exactly how the blue light was handled. This situation is found all along the manuscript.
Author Response
Thank you for your valuable comments. According to your suggestions, we have made the following revisions to this paper:
- For the language problem you pointed out, we have made modifications in the final version after polishing (Please see the attachment).
- About “The mechanism about the effect of blue light exposure on the health is not inadequate”. What this means is that the mechanisms underlying neuropathy induced by blue light remain unclear.
- My experimental design method was not clearly stated in our previous manuscript. Our experimental design was as follows: firstly, After five days of adaptation, 36 male SD rats were randomly divided into control and BLS groups according to body weight (n =18/group). Secondly, According to the requirements for the environment and housing facilities of experimental animals (GB 149252010, China), the cage containing the control subjects was raised from 10:00 to 20:00 under a 10 h light/ 14 h dark cycle (200 L × white light). BLS was supplemented with blue light for 4 hours from 6:00 to 10:00 before given white light, and the irradiance of blue light was the same as that of the control group to exclude the influence of different light energies. Finally, BLS group after 4 hours of blue light exposure, simultaneously provide white light to the control group at the same time to enter a sleep period. The experiment was conducted for 12 weeks. RNA m6A modifications and different mRNA transcriptome profiles were observed in the LHb.
- We have also corrected the experimental design drawing in the figure.
As shown in the figure:
Thank you again for your comments and kindly pointing out the various problems in the manuscript, which helped me a lot.

Reviewer 2 Report
This paper is well written and deals with an important topic. While several hypotheses about the harmful effects of blue light on mental and physical health have been suggested, there is insufficient evidence about such effects on, particularly, the lateral habenular area, which is closely related to the GABAergic system.
This paper has numerous implications not only for future basic studies but also for clinical studies with human subjects.
Author Response
Thank you for your valuable comments.
Round 2
Reviewer 1 Report
Our remaks have been completed correctly. With have no further questions.